# Research on Key Genes for Flowering of *Bambusaoldhamii* Under Introduced Cultivation Conditions

**DOI:** 10.3390/genes16070811

**Published:** 2025-07-11

**Authors:** Shanwen Ye, Xuhui Wei, Jiamei Chen, Suzhen Luo, Tingguo Jiang, Jie Yang, Rong Zheng, Shuanglin Chen

**Affiliations:** 1Fujian Academy of Forestry Sciences, Fuzhou 350000, China; yeshanwen_faf@163.com (S.Y.); maychen0805@126.com (J.C.); jiangtingguo2025@163.com (T.J.); jiey179@163.com (J.Y.); 2Jinshan Campus of Fujian Agriculture and Forestry University, Fuzhou 350002, China; 17847699436@163.com; 3Youxi State-Owned Forest Farm of Fujian Province, Sanming 365100, China; luozi7208@163.com; 4Institute of Subtropical Forestry, Chinese Academy of Forestry, Hangzhou 311400, China

**Keywords:** introduced cultivation, *Bambusaoldhamii*, flowering, key genes, environmental stresses

## Abstract

Background: *Bambusaoldhamii* is an important economic bamboo species. However, flowering occurred after its introduction and cultivation, resulting in damage to the economy of bamboo forests. Currently, the molecular mechanism of flowering induced by introduction stress is still unclear. This study systematically explored the key genes and regulatory pathways of flowering in *Bambusaoldhamii* under introduction stress through field experiments combined with transcriptome sequencing and weighted gene co-expression network analysis (WGCNA), with the aim of providing a basis for flower-resistant cultivation and molecular breeding of bamboo. Results: The study conducted transcriptome sequencing on flowering and non-flowering *Bambusaoldhamii* bamboo introduced from Youxi, Fujian Province for 2 years, constructed a reference transcriptome containing 213,747 Unigenes, and screened out 36,800–42,980 significantly differentially expressed genes (FDR < 0.05). The results indicated that the photosensitive gene *CRY* and the temperature response gene *COR413-PM* were significantly upregulated in the flowering group; the expression level of the heavy metal detoxification gene *MT3* increased by 27.77 times, combined with the upregulation of the symbiotic signaling gene *NIN*. WGCNA analysis showed that the expression level of the flower meristem determination gene *AP1/CAL/FUL* in the flowering group was 90.38 times that of the control group. Moreover, its expression is regulated by the cascade synergy of *CRY*-*HRE/RAP2-12*-*COR413-PM* signals. Conclusions: This study clarifies for the first time that the stress of introducing *Bambusaoldhamii* species activates the triad pathways of photo-temperature signal perception (*CRY*/*COR413-PM*), heavy metal detoxification (*MT3*), and symbiotic regulation (*NIN*), collaboratively driving the *AP1*/*CAL*/*FUL* gene expression network and ultimately triggering the flowering process.

## 1. Introduction

The study of flowering in bamboo plants is a complex and important field, and the flowering phenomenon often occurs during the growth process of bamboo plants [1]. Especially for bamboo species with high economic value such as *Bambusaoldhamii*, if flowering occurs after introduction, it not only affects the growth and stability of bamboo forests but may also lead to large-scale death of bamboo forests, causing a serious impact on the ecosystem and the bamboo industry. At present, the research on the flowering of bamboo usually focuses on the flowering of bamboo in mature bamboo forest stands [2,3,4,5], and there are no systematic research reports on the flowering phenomenon of bamboo caused by transplanting.

Although the flowering of bamboo is regarded as a normal physiological phenomenon, its pattern and timing are unique. Unlike many plants, bamboo usually shows the characteristic of group flowering when it flowers; that is, bamboo of the same species will flower collectively at the same time [6,7,8]. This phenomenon might be a strategy formed by bamboo during its evolution. Bamboo can enhance pollination efficiency and increase seed competitiveness through simultaneous flowering, thereby gaining a survival advantage in environmental changes [6,9,10]. Furthermore, bamboo with similar genes can coexist in time and space, forming a continuous clonal lineage. This age structure may eventually lead to intermittent or partial flowering of bamboo [11,12]. A large number of transcriptional and epigenomic analyses have shown that the flowering of bamboo is regulated by internal genetic factors [13,14,15,16]. When bamboo reaches a certain physiological state or mature stage, it will enter the flowering period [17,18,19]. For example, the FBH family is composed of multiple transcription factors [20], which can interact with *miRNA319*(*miR319*) -sensitive genes. This mechanism helps regulate the development process of flowers [21,21,22]. While *EDF1* and *EDF2* play a negative regulatory role in the development of flowers, *EDF1/2* can negatively regulate the genes related to the ethylene response, thereby affecting the senescence and flower drop process of flowers [23]. On the other hand, the *AP1/CAL/FUL* gene is a member of the *MADEs-box* gene family and was first identified in *Arabidopsis thaliana*. Its expression is closely related to the development of reproductive structures and plays a role in promoting flower formation and ensuring the correct arrangement and function of flower organs [24,25,26].

Although the fundamental cause of flowering is genetic factors, the plant stress response caused by introduction may prompt them to flower earlier [27,28,29]. The common physiological reactions of *Bambusaoldhamii* after transplantation are characterized by plant wilting, root damage, and the slowdown of photosynthesis [30]. During the introduction process, the plants need to be pruned, but this will lead to a decline in their nutrient absorption capacity and increase the risk of fungal infection [31]. Photosynthesis is an important external factor affecting the flowering of bamboo. Bamboo usually initiates its reproductive stage under specific light conditions. Studies have found that changes in sunlight duration and intensity directly affect the growth and flowering cycle of bamboo [9,32,33], and changes in the expression level of the *cry* gene under different light conditions can lead to significant differences in photosynthetic efficiency [34]. During the transplanting process, the root systems of plants are often severely damaged. The root system is the main part for plants to absorb water and nutrients. When the root system is cut or damaged during excavation and replanting, this process may cause the plant to be unable to effectively absorb the required water and nutrients in the short term, thereby affecting the growth rate and health status of the plant. For example, the expression of the *chalcone isomerase* (*CHI*) gene promotes the synthesis of flavonoids and other antioxidant substances, which can help plants resist oxidative stress caused by drought. Under drought conditions, the reactive oxygen species (*ROS*) produced in plants will increase. *ROS* can regulate the synthesis and signal transduction of growth hormones, thereby affecting the flowering time and flower formation [35,36,37,38]. Meanwhile, after transplanting, the root systems of plants need to re-establish their connection with the soil environment. The ability to obtain nutrients from the soil affects the growth conditions of bamboo. For example, the *NIN* gene encodes a transcription factor and belongs to the *RWP-RK* gene family. This type of transcription factor plays a key role in cell growth, the development of plants, and their responses to environmental stimuli [39,40,41]. The *NIN* gene is closely related to the nitrogen use efficiency of plants, especially in the signal transduction process related to nitrogen sources (such as nitrate) [42,43]. Poor or polluted soil may lead to poor growth of bamboo, thereby activating the flowering physiological process [9,44,45,46].

*Bambusaoldhamii*, as an economic bamboo species that is easy to plant and has high benefits, has been widely introduced and cultivated. However, the flowering phenomenon caused by the introduction is a complex process involving multiple physiological, morphological, and biochemical mechanisms, and there is no targeted research on it at present. To complement these previous studies, we designed an experimental method. The experiment collected *Bambusaoldhamii* from two different source areas for transplantation. Two years later, a combination of field sampling and molecular analysis was adopted for flowering bamboo. This study aims to reveal the key molecular mechanisms of the flowering phenomenon of *Bambusaoldhamii* under introduced cultivation conditions in order to address the economic losses caused thereby. The signal cascade pathway (*CRY-HRE*/*RAP2-12-COR413-PM*) of flowering induced by environmental stress (light/heavy metal/nutrition) was revealed for the first time through multi-omics analysis. The core regulatory role of *AP1/CAL/FUL* as a determinant of flower meristem establishes an “environmental signal-gene network” regulatory model to provide molecular targets for bamboo flower-resistant breeding. This achievement fills the research gap on the flowering mechanism of bamboo under introduction stress and theoretically expands the understanding of the flowering mechanism of bamboo.

## 2. Methods

### 2.1. Plant Materials

The experimental site is located in Youxi, Sanming City, Fujian Province (117°48′30″ to 118°40″ east longitude, 25°50′36″ to 26°26′30″ north latitude), with an average annual temperature of 19.2 °C, an average annual precipitation of about 1600 mm, and an average annual sunshine duration of approximately 1800 h. All the experimental Bambusaoldhamii were native to Shanghang (116°15′50″ to 116°56′47″ east longitude and 24°46′02″ to 25°27′47″ north latitude) and Fu′an (119°23′ to 119°51′ east longitude and 26°41′ to 27°24′ north latitude), and they were introduced and planted in 2022 in the form of rootstock bamboo. Two years after transplanting, in 2024, the flowering Bambusaoldhamii plants (FSH) from the sampling site in Shanghang, as well as the non-flowering Bambusaoldhamii plants (UFSH) from the sample sites in Shanghang, the non-flowering Bambusaoldhamii plants (UFFA) from the sample sites in Fu′an, the introduced Bambusaoldhamii plants in Shanghang (SH), and the introduced Bambusaoldhamii plants in Fu′an (FA) were selected as control plants for sampling. The growth of Bambusaoldhamii in this experimental field shows a typical clumping trait. The culms of healthy plants without flowering are about 6–8 m tall, and the ground diameter is about 4–6 cm (Figure 1). When the flowering plants were sampled, typical panicles were clearly formed at the top shoots and lateral branches, and the vegetative growth was basically stagnant. The top shoots of the unflowering plants maintain a normal vegetative bud form and grow vigorously.

### 2.2. RNA-Seq Sequencing and Transcriptome Assembly

This study employed a parameter-free transcriptome; the samples were extracted by ethanol precipitation method and CTAB-PBIOZOL. The successfully extracted RNA was added to 50 µL of DEPC-treated water for dissolution. Subsequently, the total RNA was identified and quantified using the Qubit fluorescence quantifier and the Qsep400 high-throughput biological fragment analyzer. Eukaryotic transcriptome mRNA is enriched with mRNA with polyA tails through Oligo(dT) magnetic beads. Then, fragmentation buffer was added to break the RNA into short fragments. Using the short RNA fragments as templates, the first strand of cDNA was synthesized using six-base random hexamers. Then, buffer solution, dNTPs (dTTP, dATP, dGTP, and dCTP), and DNA polymerase were added to synthesize two-stranded cDNA. Subsequently, double-stranded cDNA was purified using DNA purification magnetic beads. The purified double-stranded cDNA was subjected to end repair, an A tail was added, and the sequencing connector was connected. Then, the fragment size was selected using DNA purification magnetic beads. Finally, PCR enrichment was performed to obtain the final cDNA library. After the library inspection is qualified, different libraries are pooled according to the target offline data volume and sequenced on the Illumina platform. High-quality reads were obtained after filtering the original sequencing data. The transcript sequence of the species was obtained through Trinity splicing. Then, corset was used to remove redundancy to obtain the Unigene sequence. Finally, the high-quality reads were compared with the de-redundant transcript, and the expression level of the gene was calculated. Finally, calculate the differentially expressed genes in the samples of different groups, and conduct annotation and enrichment analysis on the differentially expressed genes.

### 2.3. Analysis of Differential Expressed Genes

The expression level of the transcript was calculated using RSEM v1.3.1 software, and then the FPKM of each transcript was calculated based on the length of the transcript. Differential expression analysis between sample groups was conducted using DESeq2 to obtain the set of differentially expressed genes between two biological conditions. The *p*-value was corrected using the Benjamini and Hochberg methods. The corrected *p* value and |log_2_ fold change| were used as the thresholds for significant differential expression. The screening conditions for differentially expressed genes are |log_2_ Fold Change| ≥ 1 and FDR < 0.05. The redundant transcript sequences were compared with the KEGG, NR, Swiss-Prot, GO, COG/KOG, and Trembl databases using DIAMOND v2.0.9 software, and the amino acid sequences were compared with the Pfam database using HMMER 3.2 software, thereby obtaining the annotation information of the seven major databases of the transcripts. Enrichment analysis was conducted based on hypergeometric tests. For KEGG, hypergeometric distribution tests were conducted in units of pathways. For GO, it is based on GO term. Meanwhile, iTAK is used for transcription factor prediction.

### 2.4. qRT-PCR Analysis

The key DEG of bamboo flower development was selected for real-time quantitative PCR(qRT-PCR) verification. The qRT-PCR reaction was analyzed using the PCR detection system (Bio-Rad, Hercules, CA, USA) and SYBR Green Master Mix (Takara, Dalian, China). In each reaction, 1 μL of cDNA template, 10 μL of 2 × SYBR Green Master Mix, and 1 μL (10 μmol/μL) of each primer were used, and water was added to expand to a final volume of 20 μL. The amplification procedure was one 95 °C cycle of 30 s, followed by 39 95 °C cycles of 5 s, TM cycles of 20 s, and 72 °C cycles of 20 s. Take Actin 1 as the internal reference gene. All qPCR detections were performed using three biological replicates and four technical replicates, and quantitative analysis was conducted using the 2^−ΔΔCT^ method. For each target gene in each treatment group, three independent biological repeat samples were used, and each biological repeat sample underwent four technical repeat experiments. The *Actin1* gene was used as the internal reference gene for expression standardization. Relative expression level analysis: The relative expression level of the target gene in each group was calculated using the 2^−ΔΔCT^ method. Statistical analysis: Based on the average expression level data of three biological replicates, the Tukey test method was used to analyze the statistical significance of the difference in target gene expression between the FSH group and the SH group. The significance mark: The same letter indicates an insignificant difference (*p* > 0.05), while different letters indicate an extremely significant difference (*p* < 0.05).

### 2.5. WGCNA Constructs the Co-Expression Module and Regulatory Network

Weighted gene co-expression network analysis was performed using WGCNA, version 1.71, with parameters mergeCutHeight = 0.25, RsquaredCut = 0.85, TOMType = “signed”, and minModuleSize = 50. The regulatory network was visualized using Cytoscape v3.10.2 software.

### 2.6. Statistical Analysis

Data analysis was conducted using the Tukey method. The significance mark: The same letter indicates an insignificant difference (*p* > 0.05), while different letters indicate an extremely significant difference (*p* < 0.05), and the error lines were expressed as ±standard error. The correlation coefficient *R* value between transcriptome data and qRT-PCR data was measured by the Pearson correlation coefficient method.

## 3. Results

### 3.1. Identification of Total Unigene During the Development of Bambusaoldhamii

In order to explore the influence mechanism of abiotic stress on *Bambusaoldhamii* under the introduction and cultivation conditions, this paper sampled the flowering samples of *Bambusaoldhamii* 2 years after introduction and cultivation and collected the control *Bambusaoldhamii* samples that did not flower. Transcriptome sequencing analysis was completed for a total of 15 samples, obtaining a total of 116.36 Gb of Clean Data. The Clean Data of each sample reached 6 Gb. The Q30 was distributed between 93.62% and 94.56%, and the GC content was between 52.58% and 54.51%, indicating that the data quality of the transcriptome was relatively high (Table 1). The length range of the sequences was 197–17,056 bp. The length of N50 in Unigenes was 1685 bp, and the length of N90 was 537 bp (Table 2). In this study, the Unigene sequence was compared with seven databases, namely KEGG, NR, Swiss-Prot, GO, COG/KOG, Trembl, and Pfam, to obtain the annotation information of Unigene. Among them, the TrEMBL database annotated 151,311 (70.95%) unigenes, while at least one database annotated 155,710 (73.1%) unigenes (Appendix A). From the perspective of the correlation heat map, the correlation among the biological replicates of the samples in the same group is relatively high, indicating good repeatability within the selected sample group (Figure 2). Furthermore, the PCA results showed that the flowering *Bambusaoldhamii* in Shanghang was significantly distinguished from other unflowering *Bambusaoldhamii* (Figure 3). According to the GO analysis (Figure 4, Appendix A), genes related to “reproduction”, “reproductive process”, and “stimulus response” were enriched. To identify the biochemical pathways, we plotted the annotation sequences in the KEGG database (Figure 5, Appendix A). Most unigenes are involved in “plant–pathogen interactions”, “protein processing in the endoplasmic reticulum”, ribosomes, and “MAPK signaling pathway—plants”. These pathway allocations provide valuable information for the study of specific biochemistry and development.

### 3.2. Screening of Differentially Expressed Genes in Introduced Cultivation and Enrichment Analysis of Their Biological Functions

The screening criteria for differentially expressed genes in this project are (|log_2_ Fold Change|≥ 1 and FDR < 0.05). By comparing FSH vs. UFFA, FSH vs. FA, FSH vs. UFSH, and FSH vs. SH in four groups, specific highly expressed genes in Shanghang flowering bamboo were screened out. A total of 36,800, 42,980, 35,790, and 39,352 differentially expressed genes were screened out, respectively. Among them, the number of upregulated genes was 19,617, 21,522, 18,436, and 19,614. The Venn diagram was used to present the intersection and specific conditions of differential genes in *Bambusaoldhamii* before and after introduction and cultivation (Figure 6). The results showed that before and after the introduction and cultivation, the number of specific highly expressed genes in Shanghang was 20,234. The results indicated that the introduction and cultivation induced transcriptome changes in the top branches of *Bambusaoldhamii*. To verify the biological functions of the differential genes for flowering of *Bambusaoldhamii* before and after introduction and cultivation, we conducted GO and KEGG enrichment analyses on the differential genes (Figure 7 and Figure 8). After removing the low-expression DEGs (FPKM < 10), all the differentially expressed genes were annotated into the GO database. In the classification of biological processes, differentially expressed genes are enriched in the regulation of responses to external stimuli (115, 4.08%), ethylene responses (113, 4.01%), and the regulation of responses to biological stimuli (110, 3.9%). In the molecular functional category, the differentially expressed genes were also enriched in monooxygenase activity (77, 2.41%), dioxide activity (69, 2.16%), and carboxylic acid transmembrane transporter activity (60, 1.88%). In the KEGG annotation results, the specifically highly expressed genes in the flowering plants of FSH g *Bambusaoldhamii* were annotated into 126 KEGG pathways. The results of KEGG enrichment analysis showed that the differentially expressed genes in the classification of environmental information processing were enriched in the plant hormone signal transduction (138, 9.56%) and MAPK-signaling pathway—plant (81, 5.61%) pathways. In the classification of genetic information processing, differentially expressed genes were significantly enriched in protein processing in ribosomes (168, 11.64%) and endoplasmic reticulum (108, 7.48%). In the classification of metabolism, differentially expressed genes were enriched in the KEGG pathways of metabolic pathways (659, 45.67%), biosynthesis of secondary metabolites (435, 30.15%), and carbon metabolism (91, 6.31%).

### 3.3. External Stimuli Regulate DEGs, Causing Bambusaoldhamii to Flower

When studying the key genes for the flowering of *Bambusaoldhamii* under introduced cultivation conditions, we analyzed the effects of different external stimuli (such as light, oxygen, temperature, water and heavy metals, etc.) on the differentially expressed genes (DEGs) of Bambusaoldhamii and explored the key roles of these genes in the flowering process of *Bambusaoldhamii* (Figure 9A). The expression levels of the light-sensing genes *CRY* and *SHW* in the flowering bamboo of the experimental group (FSH) were 10.55 and 11.18, respectively, which were significantly higher than those of the control group (SH). (4.84 and 5.37) (Figure 9B), indicating that light plays an important role in regulating the flowering process of *Bambusaoldhamii*. It may trigger the expression of a series of downstream genes by regulating the activities of cryptochrome and other photoreceptors, thereby initiating the flowering mechanism. The oxygen-sensing gene *HRE/RAP2-12* and its related genes showed significantly high expression under FSH conditions, with specific expression levels of 187.39 and 109.80 (Cluster-89802.19 and Cluster-89802.14), while those in the control group (SH) were 29.53 and 32.63, respectively (Figure 9C). Hypoxic conditions may regulate the flowering process by increasing the generation of reactive oxygen species (*ROS*) and activating oxygen-sensing genes. Furthermore, the temperature-sensing genes *COR413-PM* and *TOT3* showed significantly high expression under FSH conditions, with specific expression levels of 151.27 and 11.32, respectively, while those of the control group (SH) were 51.84 and 5.24, respectively (Figure 9D), indicating that temperature changes have a significant impact on bamboo flowering. It may regulate the expression of flowering genes by influencing the fluidity of cell membranes, protein structure, and metabolic pathways. The high expression of water and drought stress genes *SEU/SLK* and *SOS4/SNO1* under FSH conditions suggests that drought stress may regulate the expression of related genes by changing the osmotic pressure and water potential of cells, thereby affecting the flowering process.

The heavy metal detoxification genes *MT3* and *MT4* showed extremely high expression levels under FSH conditions, which were 25,764.74 and 38.46, respectively, while the expression levels (FPKM) of the control group (SH) were 927.97 and 5.07, respectively (Figure 9E). This indicates that more heavy metal stress needs to be dealt with during the flowering process of *Bambusaoldhamii*, possibly by encoding metallothionein binding and neutralizing heavy metal ions to reduce their toxic effects. It is particularly notable that the expression level of the heavy metal detoxification gene *MT3* under FSH conditions is 27.77 times that of SH, demonstrating the most significant expression difference. The symbiotic signaling pathway gene *NIN* and its related genes showed significantly high expression under FSH conditions, with specific expression levels of 15.12 (Cluster-23561.57) and 28.78 (Cluster-23561.65), while those of the control group (SH) were 6.78 and 16.34, respectively (Figure 9F). This implies that symbiotic microorganisms promote the flowering of *Bambusaoldhamii* by providing nutrients, enhancing immunity, or regulating hormone levels. The expression levels of pathogen defense genes *RIN4* and *CHIA* significantly increased under FSH conditions, which were 19.80 and 130.16, respectively, while those of the control group (SH) were 4.93 and 16.47, respectively (Figure 9G). This indicates that during the flowering process of *Bambusaoldhamii*, it is necessary to enhance its defense mechanism, possibly by encoding defense enzymes and proteins to recognize and resist pathogens, thereby protecting plant health and promoting flowering. Especially the pathogen defense gene *CHIA*, its expression level under FSH conditions is 7.90 times that of SH.

### 3.4. DEGs Related to Flower Organ Development Are Involved in the Flower Development of Bambusaoldhamii

The development of flowers during the flowering process of bamboo has always been one of the important fields in botanical research. Genes related to the development of flower organs play a key role in regulating the morphological formation, development process, and reproductive success of flowers. We screened out the differentially expressed genes related to the development of flower organs and functionally annotated and classified these genes (Figure 10A). The results show that among the genes related to flower formation, the expression levels of *FBH* family members (such as *FBH1/2/++*, *FBH1/3/++*, *FBH1/4/++*, *FBH1/5/++*, *FBH1/6/++*, and *FBH1/7/++*) under FSH conditions were 25.24, 21.41, 36.54, 13.04, 43.63, and 15.28, respectively. Significantly higher than the control group (such as 4.92, 4.14, 11.15, 1.69, 12.11, and 0.97 in SH) (Figure 10B). This indicates that members of the *FBH* gene family may be involved in regulating the temporal process of *Bambusaoldhamii* flower development, such as determining the initiation time of flower bud differentiation or the transitions at different stages of flower organ development.

The expression level of the *EDF1/2/++* gene under FSH conditions was 160.45, while that of the control group (SH) was only 1.29, showing an extremely high difference. This suggests that it may affect the formation and development of the organs of the *Bambusaoldhamii* flower by regulating the expression of downstream genes during the photoperiod signal transduction process. For example, regulating the formation of flower primordia or the differentiation direction of flower organ primordia, etc. (Figure 10C). In addition, the expression levels of *FVE* and *LD* genes under FSH conditions were also significantly higher than those in the control group, indicating the importance of the autonomous flower-promoting pathway and post-transcriptional regulation in the flowering of *Bambusaoldhamii*. By modifying the chromatin structure, these genes affect the expression of genes related to the development of flower organs, thereby regulating the development process of flower organs, such as the morphological formation and cell differentiation of organs like petals and stamens.

The expression levels of the *AP1/CAL/FUL* gene with significant specificity under FSH conditions were 290.92 (cluster-45525.0) and 338.60 (cluster-45525.47), while in the control group (SH), they were 3.22 and 32.36, respectively (Figure 10D). This shows a huge difference in the transcriptional regulation of flower bud recognition. In particular, the expression level of cluster-45525.0 is 90.38 times that of the control group, which has important biological significance. The expression changes of it in different samples indicate that it may play a key role in determining the characteristics of the flower meristematic tissue, such as determining the formation location, size of the flower meristematic tissue, and its potential for differentiation into different flower organ primordia, thereby having a decisive impact on the morphology and structure of the entire flower organ.

### 3.5. Construction of the Co-Expression Module and Regulatory Network of Flowering Genes in Bambusaoldhamii

To explore the intrinsic connection between the flowering process of *Bambusaoldhamii* and external stimuli in greater depth, we selected 15 *Bambusaoldhamii* transcriptome samples and conducted weighted gene co-expression network analysis (WGCNA) on them. This advanced analytical method can help us explore the potential interaction relationships among genes. Through WGCNA analysis, these genes were clustered into 12 co-expression modules (Figure 11). These modules have significant differences in the number of genes. Among them, the smallest module is the purple module, which contains only 416 genes. The largest module is the turquoise module, which has as many as 12,901 genes. This indicates that the genes covered by different modules may play different roles to varying degrees during the flowering process of *Bambusaoldhamii*. When conducting research on the flowering bamboo samples from Shanghang, we discovered a very interesting phenomenon. All the samples of Shanghang flowering bamboo showed a strong correlation in the blue module (Figure 12). This discovery suggests that the genes in the blue module may play a crucial role in the flowering process of *Bambusaoldhamii*.

To further analyze the interrelationships among genes in the Blue module, we screened out the network node relationships in the module (weight0.3) and then drew the network diagram using Cytoscape v3.10.2 (Figure 13). It was found that 11 specific highly expressed genes showed significant expression under the conditions of the experimental group. After further in-depth analysis of these genes, it was found that they have a close relationship with a variety of external stimulating factors.

External stimuli, as key initiating factors, can trigger corresponding changes in a series of genes. These genetic changes further stimulate the flowering of *Bambusaoldhamii*, promoting the progress of this important biological process. Specifically, in terms of external stimuli related to oxygen, the *HRE/RAP2-12* gene will undergo corresponding changes. For the external stimulus factor of temperature, the change of temperature can be perceived by the *COR413-PM* gene, which is relatively sensitive to it, converting the external temperature fluctuation into an internal signal and playing an important role in inducing the flowering of *Bambusaoldhamii*. At the level of external stimuli related to nutrition, when the external nutritional components change, the *NIN* gene will be stimulated, actively exerting its regulatory function and deeply participating in the nutritional metabolism process within *Bambusaoldhamii*. Under external stimuli related to water, when the external water conditions change, the *CHIB* gene will regulate the water balance and related physiological activities in *Bambusaoldhamii*, thereby exerting a significant regulatory effect on the key link of *Bambusaoldhamii* flowering and becoming one of the key factors determining whether *Bambusaoldhamii* flowers and affecting the quality of flowering. Furthermore, under the external stimulus of heavy metal stress, the *MT3* gene will change accordingly. By participating in processes such as chelation and transport of heavy metals, the *MT3* gene is dedicated to maintaining the relative stability of the internal environment of *Bambusaoldhamii*, and its changes indirectly affect the related physiological mechanisms of *Bambusaoldhamii* flowering. Under the specific external stimulus condition of salt stress, the *SOS4/SNO1* gene will be activated and undergo corresponding changes. Subsequently, the stress response mechanism is initiated, and the adverse effects brought by salt stress are alleviated through various pathways, such as regulating ion balance. These external stimuli affect flowering genes such as *AP1/CALI/FUL* and *EDF1/2/++* by influencing key transcription factors such as *SR45a*, *PTB,* and *RNS*. This series of molecular events ultimately constitutes the molecular mechanism that leads to the blooming of *Bambusaoldhamii*, providing important insights into the complexity of this phenomenon in a deeper understanding.

### 3.6. The Gene Expression Was Verified by qRT-PCR

In the independent biological experiments using qRT-PCR, the transcriptional regulation revealed by RNA-seq was confirmed. Seven genes were selected, professional primer design software was used, and, combined with the nucleotide sequence characteristics of the genes, highly specific primers (Appendix A) were designed. Meanwhile, Actin 1 was selected as the internal reference gene to ensure the accuracy and reliability of the experimental results. The expression of this internal reference gene is relatively stable under different experimental conditions and can effectively correct the differences between samples. After completing the primer design and sample preparation, the experiment was conducted strictly in accordance with the experimental operation process of qRT-PCR. After a series of steps such as reverse transcription, amplification, and fluorescence signal detection, the expression data of each gene were obtained. The abundance data of transcripts obtained by RNA-seq determination were compared and analyzed with the gene expression profiles revealed by qRT-PCR. The results show that there is good reproducibility between the two (Figure 14).

## 4. Discussion

Through Illumina sequencing and transcriptome detection, this study obtained high-quality gene expression data (Q30 > 93.62%, GC content 52.58–54.51%) during the development of green bamboo flowers. This data quality is highly consistent with the transcriptome analysis (Q30-92%) in the Bambusa tulda flowering study by Chakraborty et al. (2021) [5], but in this study, through more rigorous sample screening (flowering vs. the non-flowering group) and the repetitive design (three biological replicates), reliability was significantly enhanced. The latest research, such as Liu et al. (2023) [4], demonstrated similar high GC content characteristics (53–55%) in Bambusa tuldoides, indicating the conservation of the bamboo flowering transcriptome. This is mutually corroborated with the dynamic change pattern of gene expression observed by Fan et al. (2022) [47] in the transcriptome study of the flowering of Bambulosa. However, the PCA analysis of this study (Figure 3) for the first time revealed a clear distinction in the introduced cultivation samples, making up for the previous limitation of only focusing on the flowering of mature bamboo forests.

By comparing the four groups (FSH vs. UFFA, FSH vs. FA, FSH vs. UFSH, and FSH vs. SH), a large number of differentially expressed genes (DEGs) were screened out, and GO and KEGG functional enrichment analyses were conducted. The results showed that the differentially expressed genes were significantly enriched in environmental adaptability and metabolic pathways, which was consistent with the findings of Ge et al. (2016) [13] in the study of *Phyllostachys edulis* flowering (enriched in “biosynthesis of secondary metabolites”). However, this study further identified a high enrichment of “plant hormone signal transduction” in the KEGG pathway (138 genes, 9.56%), highlighting the unique role of hormone regulation under introduction stress and surpassing the previous paradigm that only focused on the photopod pathway [9].

The role of external stimuli in the flowering mechanism of plants has received increasing attention. Experiments have found that external factors such as light, oxygen, temperature, moisture, and heavy metals jointly affect the flowering process of *Bambusaoldhamii* by regulating specific differentially expressed genes (DEGs). The light-sensing genes *CRY* and *SHW* were significantly highly expressed in the experimental group, indicating that light plays a key role in the flowering of *Bambusaoldhamii*. Some studies have pointed out that under strong light irradiation, the expression level of photosensitive genes in bamboo plants significantly increases, thereby regulating the growth and flowering mechanisms of bamboo [32,48]. It is worth noting that the ROS regulatory mechanism of the *HRE/RAP2-12* gene is consistent with the hypoxia-induced flowering model [49,50]. In addition, this study found that the expression difference of COR413-PM in temperature response was more significant (2.92 times), which is consistent with the mechanism by which temperature-sensing genes affect flowering time through membrane fluidity [51,52,53]. However, this study was the first to confirm in *Bambusaoldhamii* the synergistic effect of elevated ROS levels and the *MT3* gene (27.77 times upregulated), providing molecular evidence for the “stress-induced flowering” theory.

During the flowering process of green bamboo, genes related to the development of floral organs, such as *AP1/CAL/FUL* (expression level 290.92 vs. 3.22, difference 90.38 times), play a core role. This is consistent with the finding of Ge et al. (2016) [13] in *Phyllostachys edulis* (AP1 was upregulated by approximately 50 times). Relevant studies have shown that the APETALA1 (*AP1*) and CAULIFLOWER (*CAL*) genes play a key role in the development of flower organs in the bamboo, and significant changes in their expression levels are closely related to flower bud formation [54,55]. Through weighted gene co-expression network analysis (WGCNA), this study constructed the co-expression module and regulatory network of the flowering gene of *Bambusaoldhamii*. The blue module (12,901 genes) is strongly associated with the flowering process of green bamboo. This method is similar to the application of De Silva et al. (2022) [56] in the embryonic development of Arabidopsis thaliana. However, in this study, *HRE/RAP2-12* and COR413-PM were identified as core transcription factors in bamboo for the first time. Their regulatory networks are more complex and involve the integration of multiple external stimuli. In the future, it is necessary to combine single-cell redemption technology to deepen the understanding of the dynamic nature of modules.

## 5. Conclusions

This study found that abiotic stresses such as light, temperature, water imbalance, and heavy metal pollution during the introduction and cultivation of *Bambusaoldhamii* can activate the *CRY-HRE/RAP2-12-COR413-PM-MT3-AP1/CAL/FUL* gene chain network, thereby inducing the flowering of green bamboo. This mechanism is of great significance for bamboo production in the background of global warming. High temperature and strong light, as the key stress factors triggering flowering, can extend the economic lifespan of bamboo forests by screening the low-expression genotypes of *COR413-PM/CRY* or creating light-temperature-insensitive germplasms through gene editing. The synergistic activation of the heavy metal detoxification gene *MT3* and the symbiotic nutrient gene *NIN* reveals the interaction mechanism between oxidative stress and nutritional imbalance. Before the introduction and cultivation of *Bambusaoldhamii*, soil remediation and nutrient management optimization are necessary to block flowering signal transduction. At the same time, a coordinated monitoring system of light, temperature, and water should be established. Through intelligent irrigation and shading measures, environmental stress should be controlled below the flowering threshold to ensure the vegetative growth of bamboo forests.

## Figures and Tables

**Figure 1 genes-16-00811-f001:**
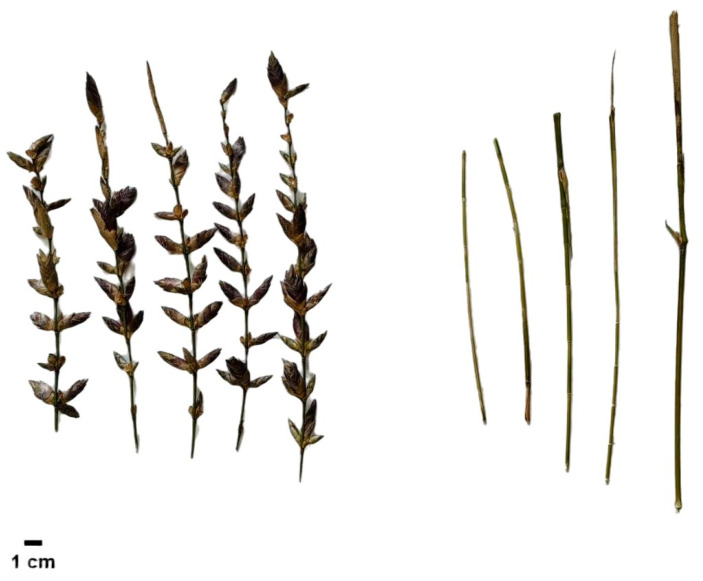
*Bambusaoldhamii* flowering bamboo (left) and non-flowering bamboo (right) samples.

**Figure 2 genes-16-00811-f002:**
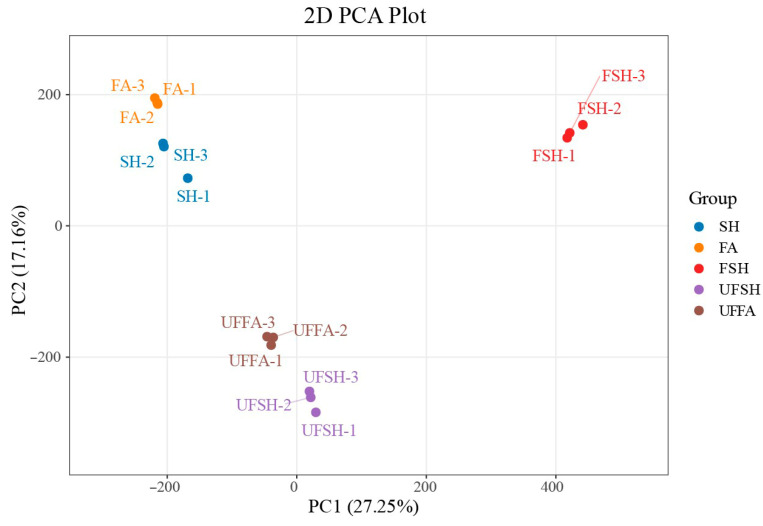
Principal component analysis of *Bambusaoldhamii* samples.

**Figure 3 genes-16-00811-f003:**
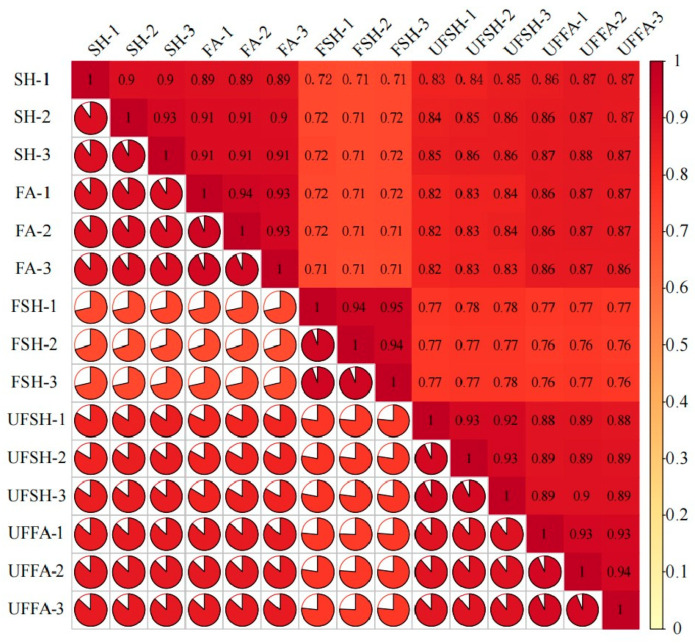
Correlation analysis of *Bambusaoldhamii* samples.

**Figure 4 genes-16-00811-f004:**
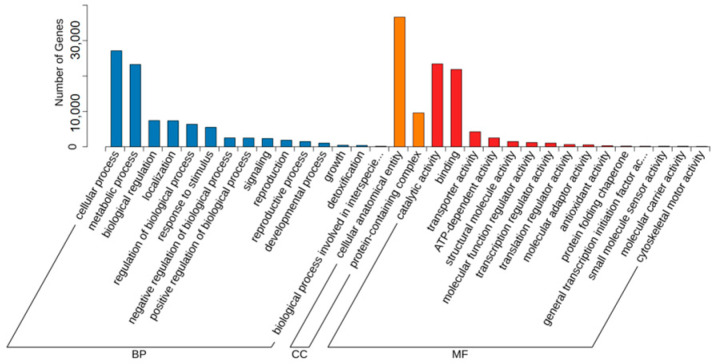
Total unigene GO classification.

**Figure 5 genes-16-00811-f005:**
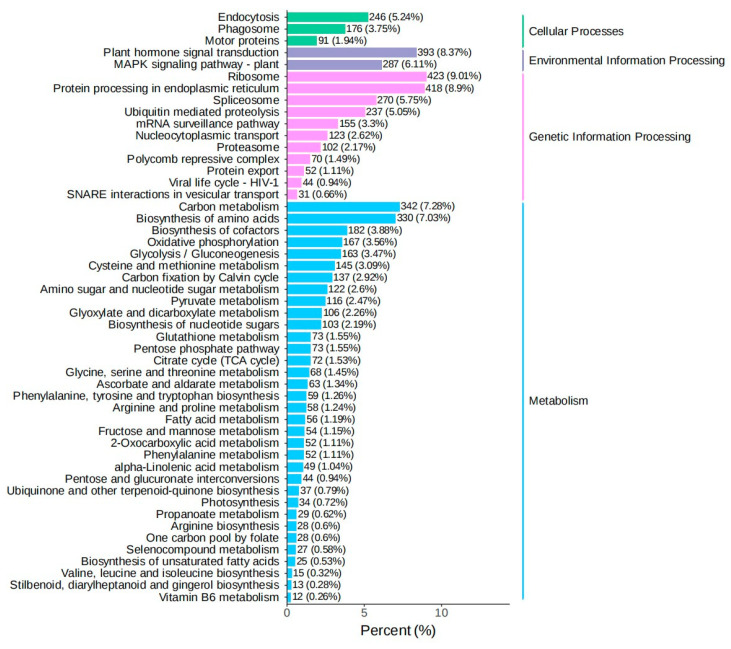
Total unigene KEGG classification.

**Figure 6 genes-16-00811-f006:**
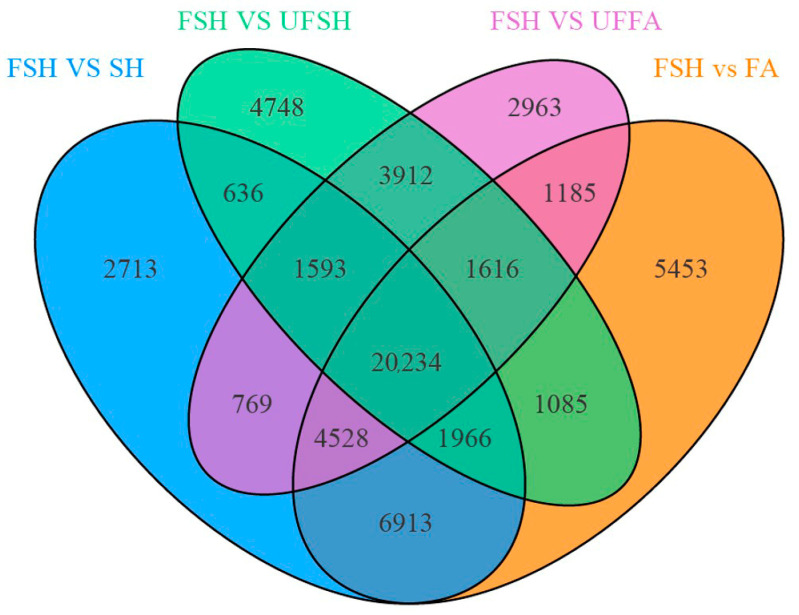
Veen map of differential genes in *Bambusaoldhamii* before and after introduction and cultivation.

**Figure 7 genes-16-00811-f007:**
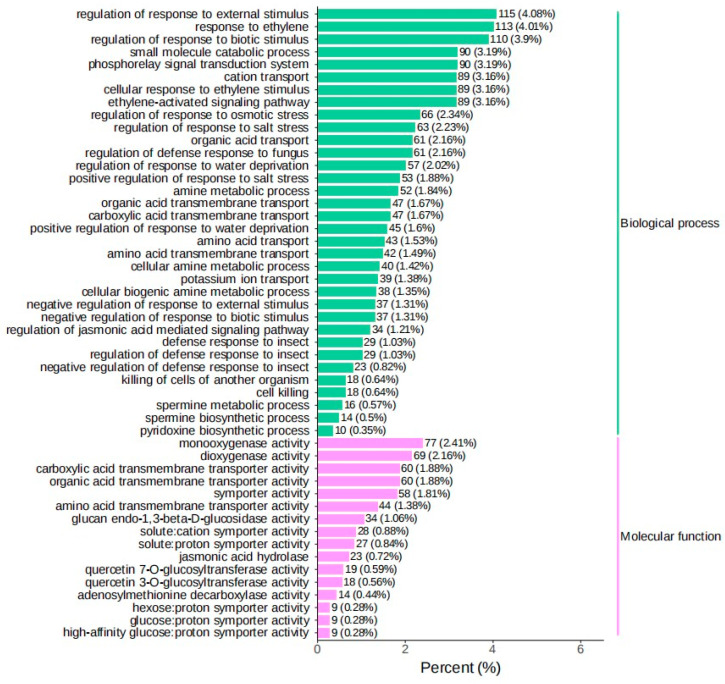
Differential genes GO enrichment analysis.

**Figure 8 genes-16-00811-f008:**
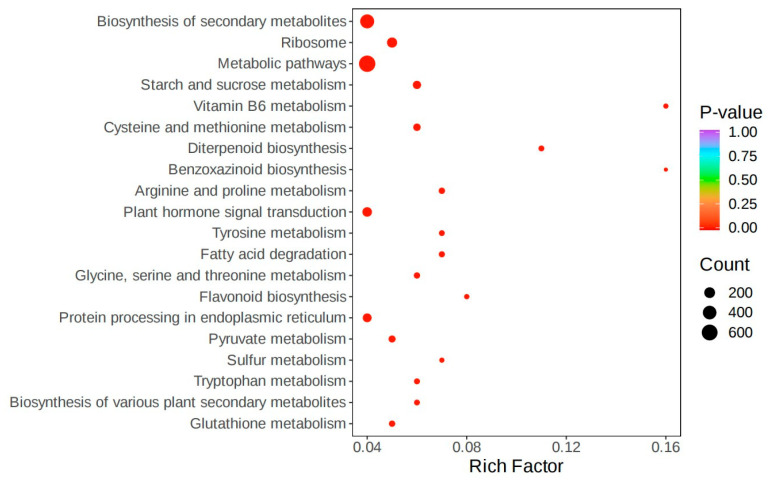
Differential genes KEGG enrichment analysis.

**Figure 9 genes-16-00811-f009:**
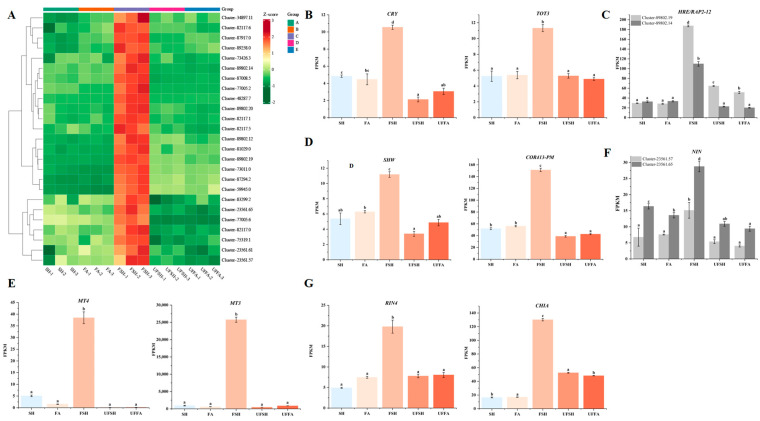
**Heat map and FPKM of key genes related to external stimuli in different samples**. (Error line: ±standard error, represents the degree of data dispersion; Significance mark: The same letter indicates an insignificant difference (*p* > 0.05), while different letters indicate a significant difference (*p* < 0.05)). (**A**) Heat maps of key gene expression associated with external stimuli in different samples; (**B**) FPKM of the light-sensing genes *CRY* and *SHW* among different samples; (**C**) FPKM of the oxygen-sensing gene *HRE/RAP2-12* among different samples; (**D**) FPKM of the temperature-sensing genes *COR413-PM* and *TOT3* among different samples; (**E**) FPKM of heavy metal detoxification genes *MT3* and *MT4* among different samples; (**F**) FPKM of the symbiotic signaling pathway gene *NIN* among different samples; (**G**) The FPKM of pathogen defense genes *RIN4* and *CHIA* among different samples.

**Figure 10 genes-16-00811-f010:**
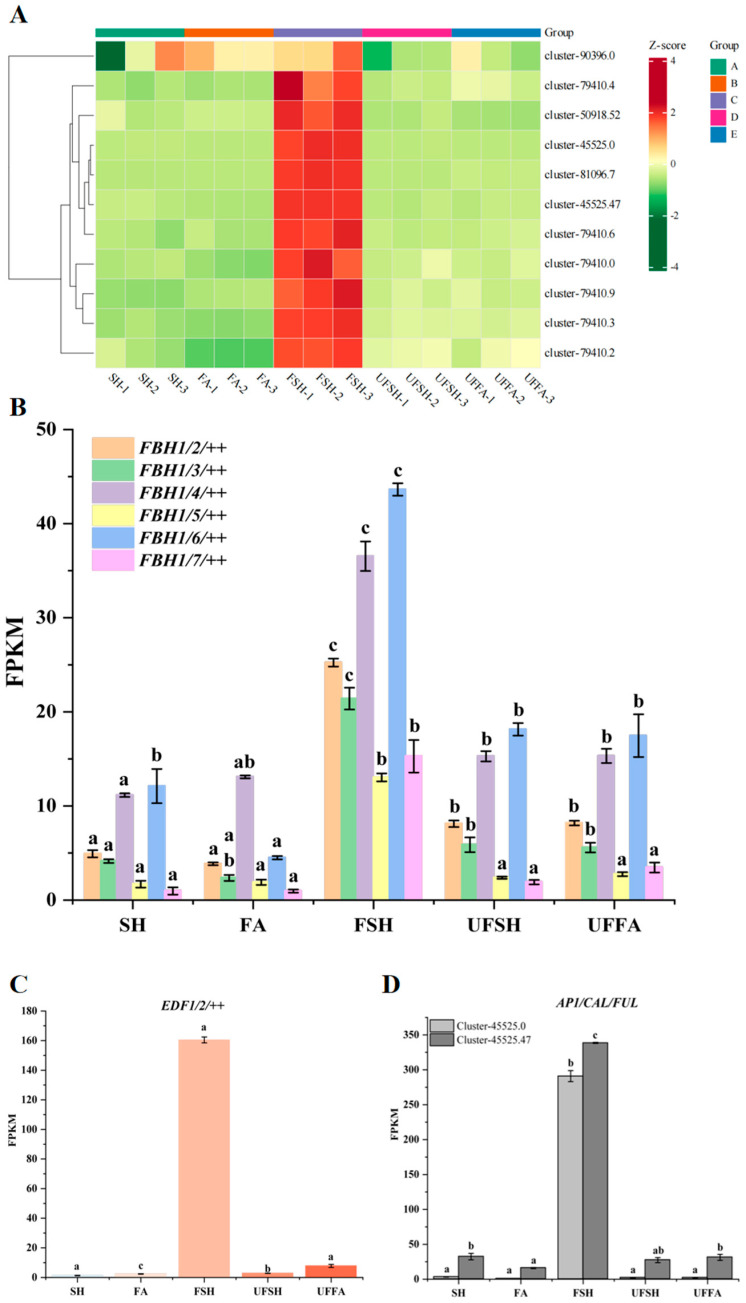
**Shows heat maps of key gene expression related to floral organ development and their FPKM in different samples** (Error line: ±standard error, represents the degree of data dispersion; Significance mark: The same letter indicates an insignificant difference (*p* > 0.05), while different letters indicate an significant difference (*p* < 0.05)). (**A**) Heat maps of key gene expression related to flower organ development in different samples; (**B**) The FPKM of *FBH* family members among different samples; (**C**) FPKM of the *EDF1/2/++* gene among different samples; (**D**) The FPKM of the *AP1/CAL/FUL* gene among different samples.

**Figure 11 genes-16-00811-f011:**
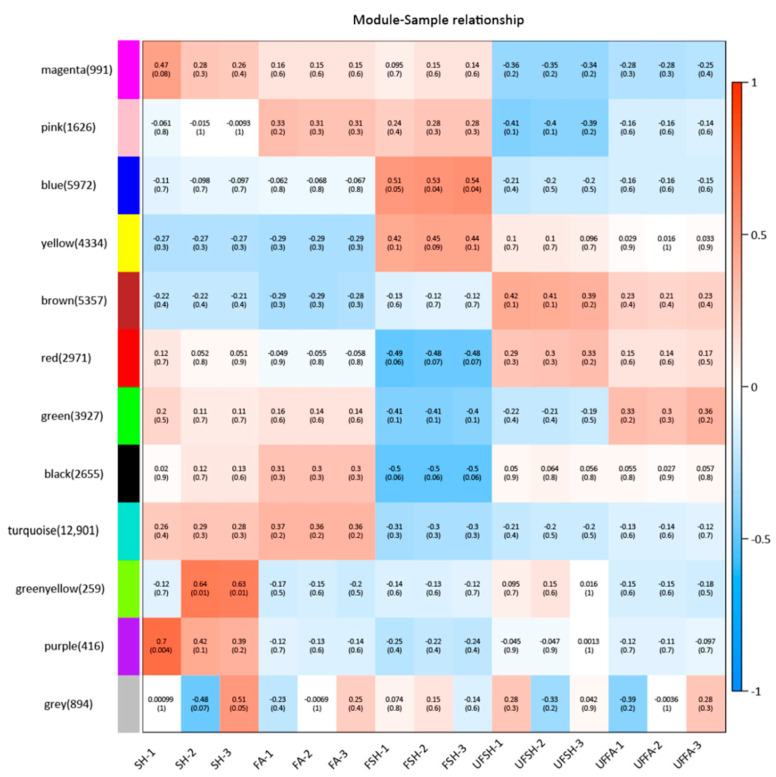
Weighted gene co-expression network analysis of each sample.

**Figure 12 genes-16-00811-f012:**
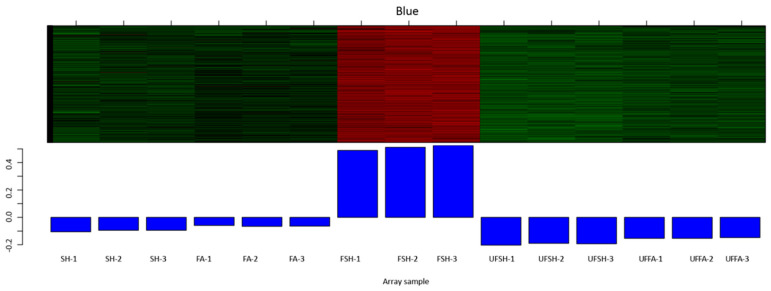
The degree of gene expression in the blue module of each sample. Red represents positive correlation and green represents negative correlation.

**Figure 13 genes-16-00811-f013:**
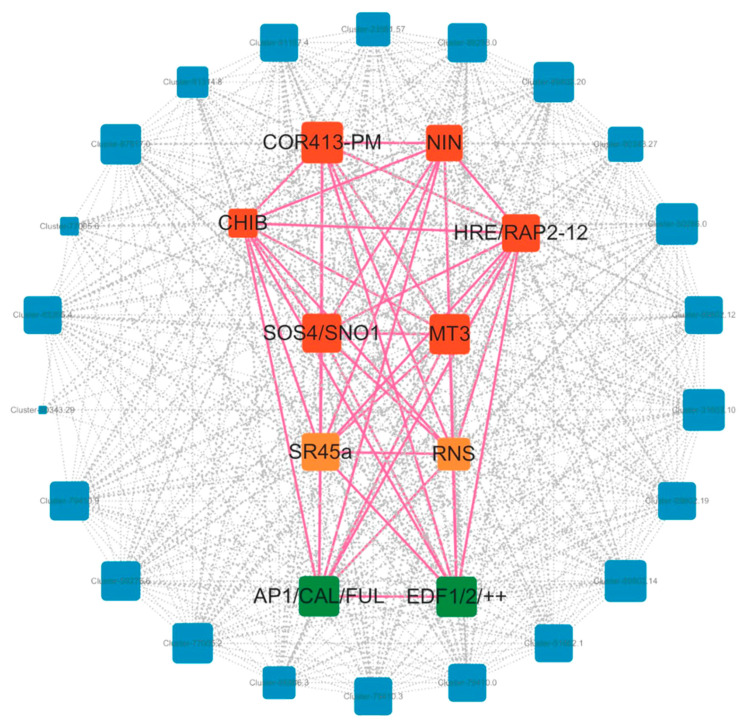
Expression network map of 10 specific high-expression genes.

**Figure 14 genes-16-00811-f014:**
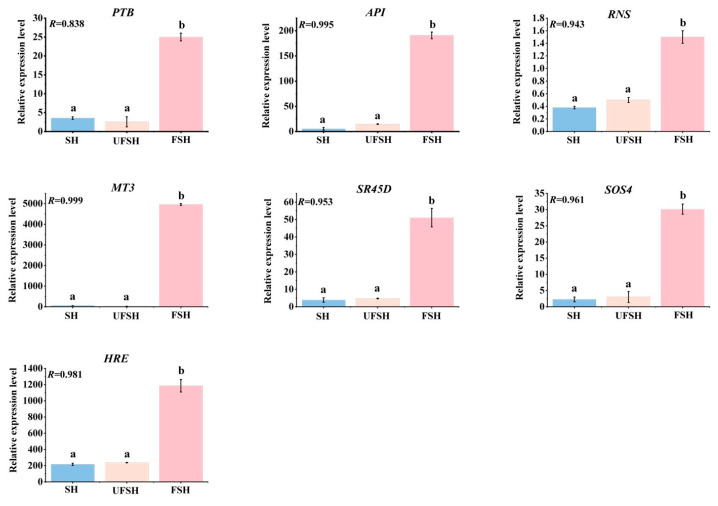
**Quantitative real-time RT-PCR confirmation of seven candidate genes**. (Error line: ±standard error, represents the degree of data dispersion; Significance mark: The same letter indicates an insignificant difference (*p* > 0.05), while different letters indicate a significant difference (*p* < 0.05); the correlation coefficient *R* between transcriptome data and qRT-PCR data).

**Table 1 genes-16-00811-t001:** The analysis of data output quality.

Sample	Raw Reads	Clean Reads	Clean Base (G)	Error Rate (%)	Q20 (%)	Q30 (%)	GC Content (%)
FA-1	62,102,718	57,080,886	8.56	0.01	97.86	93.62	53.5
FA-2	58,318,738	53,194,242	7.98	0.01	98.01	94.07	53.42
FA-3	53,872,404	48,781,900	7.32	0.01	97.96	93.92	53.24
UFFA-1	54,141,882	50,014,552	7.5	0.01	98.04	94.11	53.56
UFFA-2	57,137,426	51,287,204	7.69	0.01	98.15	94.44	53.95
UFFA-3	56,744,142	52,810,016	7.92	0.01	97.91	93.75	53.91
SH-1	58,328,184	53,256,984	7.99	0.01	97.85	93.65	54.51
SH-2	48,324,260	43,677,018	6.55	0.01	97.9	93.78	53.48
SH-3	55,391,694	50,735,326	7.61	0.01	97.88	93.69	53.38
FSH-1	66,474,268	60,477,054	9.07	0.01	98.2	94.56	52.7
FSH-2	64,482,610	57,003,080	8.55	0.01	98.1	94.28	52.58
FSH-3	58,368,886	52,412,646	7.86	0.01	98.04	94.11	52.97
UFSH-1	53,820,998	48,669,652	7.3	0.01	98.01	94.03	52.81
UFSH-2	56,564,710	50,818,384	7.62	0.01	97.94	93.8	53.08
UFSH-3	50,249,878	45,604,840	6.84	0.01	97.89	93.64	53.65

**Table 2 genes-16-00811-t002:** Assembly result statistics.

Typle	Number	Mean Length	N 50	N 90
Transcript	377,416	945	1492	390
Unigene	213,273	1174	1685	537

## Data Availability

The original contributions presented in this study are included in the article/Appendix A. Further inquiries can be directed to the corresponding authors.

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
