# Peer review of "Research on Key Genes for Flowering of Bambusaoldhamii Under Introduced Cultivation Conditions"

_genes, 2025, doi:10.3390/genes16070811_

Round 1

Reviewer 1 Report

Comments and Suggestions for Authors

The authors reports that the stress of introducing Bambusaoldhamii species activates the triad pathways of photo-temperature signal perception (CRY/COR413-PM), heavy metal detoxification (MT3), and symbiotic regulation (NIN), collaboratively driving the AP1/CAL/FUL gene expression network and ultimately triggering the flowering process. These results are novel and valuable to understand the flowering of Bambusaoldhamii. However, there some issues in this manuscript as described below.

  1. Abstract is too long. It should be shortened to about 2/3 of its original length.

  1. Significance is mentioned in many places in the text. However, there is no description of the statistical analysis. A section should be added to the Methods section to describe the statistical analysis. However, if Duncan's multiple range test is used, it is necessary to use another statistical analysis, since Duncan's multiple test has been statistically rejected.

  1. The Methods section should explain what the groups "FA", "UFFA", "SH", "FSH" and "UFSH" refer to.

  1. In many of the graphs shown in Figure 9 and Figure 10, the bars are connected by straight lines. These lines seem unnecessary. If necessary, the reason should be explained in the captions.

  1. In Figures 9 and 10, the captions should explain what the error bars represent.

  1. The letters in Figure 14 need to be enlarged. Also, the names of the genes should be listed for each graph. In addition, the captions should explain what the error bars represent.

Overall, I think that this manuscript has valuable findings.

Author Response

1. Point-by-point response to Comments and Suggestions for Authors

Comments 1: Abstract is too long. It should be shortened to about 2/3 of its original length.

Response 1: Thank you for pointing this out. We agree with this comment. Therefore, on the premise of ensuring the original meaning, we reduced the abstract as needed (Page 2, paragraphs 13-34).

Comments 2: Significance is mentioned in many places in the text. However, there is no description of the statistical analysis. A section should be added to the Methods section to describe the statistical analysis. However, if Duncan's multiple range test is used, it is necessary to use another statistical analysis, since Duncan's multiple test has been statistically rejected.

Response 2: Thank you for pointing this out. We have supplemented the description of statistical analysis in the methods section as required and conducted data analysis using the Tukey test method. The significance mark: the same letter indicates an insignificant difference (P>0.05), while different letters indicate an extremely significant difference (P<0.05)..The correlation coefficient R value between transcriptome data and qRT-PCR data was measured by the Pearson correlation coefficient method.(Page 8, lines 167-171)

Comments 3: The Methods section should explain what the groups "FA", "UFFA", "SH", "FSH" and "UFSH" refer to.

Response 3: Thank you for pointing this out. We have explained "FA", "UFFA", "SH", "FSH" and "UFSH" as required at the plant materials section of the method section (Page 5, lines 107-117).

Comments 4: In many of the graphs shown in Figure 9 and Figure 10, the bars are connected by straight lines. These lines seem unnecessary. If necessary, the reason should be explained in the captions.

Response 4: Thank you for pointing this out. These lines were originally intended to serve as trend lines to represent the changing trends of these genes in various treatments. According to the suggestions, these lines seemed unimportant, so the trend lines in these graphs were deleted.(Figures 9- 10 in the "ALL figure" folder)

Comments 5:In Figures 9 and 10, the captions should explain what the error bars represent.

Response 5: Thank you for pointing this out. The error line in Figure 9-10 is expressed as ± standard error, which represents the degree of dispersion of the data. We have explained it in the title as required. (Figure captions in ALL figure)

Comments 6:The letters in Figure 14 need to be enlarged. Also, the names of the genes should be listed for each graph. In addition, the captions should explain what the error bars represent.

Response 5: Thank you for pointing this out. We have enlarged the letters in Figure 14 as required and listed the names of the genes in the figure. The error line of the title is also explained in the title of the figure. The error line is ± standard error, representing the degree of dispersion of the data.(figure 14 and Figure captions in ALL Figures)

Reviewer 2 Report

Comments and Suggestions for Authors

Manuscript genes-3723243 “Research on Key Genes for Flowering of Bambusaoldhamii Un-2 der Introduced Cultivation Conditions” presents and interesting work about regulation of flowering in bamboo.

However instead of the great work performed and the amount of analyzed data a major revision is need.

Objective of the work (lines 104-115) must be summarized indicating main contributions of the new study

Plant material (lines 118-127) section must be completed including morphological description and the pedigree and origin of the assayed bamboo genotype.

RNA-seq protocol (153-167) should be explained indicating statistical analysis, bioinformatic analysis, reference genome, read length, biological and technical replications, etc.

qPCR protocol (lines 166-178) should be explained indicating statistical analysis, and biological and technical replications.

Figures 9a, b, c, d, f, g must be merged in a single Figure.

Figures 10a, b, c, dg must be merged in a single Figure.

In Figure 14 authors should be incorporated the correlation coefficient between qPCR and RNA-Seq data in the assayed genes.

Discussion section is limited. Authors must discuss the obtained results with previous data obtained by differ authors.

Conclusions section (lines 504-514) must be completed with the main implications of the obtained results at bamboo production level in the context of global warming.

Author Response

1. Point-by-point response to Comments and Suggestions for Authors

Comments 1: Objective of the work (lines 104-115) must be summarized indicating main contributions of the new study

Response 1: Thank you for pointing this out.We have described the main contributions of the new research (page 5, lines 99-108).

Comments 2: Plant material (lines 118-127) section must be completed including morphological description and the pedigree and origin of the assayed bamboo genotype.

Response 2: Thank you for pointing this out.We have supplemented the morphological description, pedigree and the origin of the determined bamboo genotypes in the plant materials as required (pages 6, 110-127).

Comments 3: RNA-seq protocol (153-167) should be explained indicating statistical analysis, bioinformatic analysis, reference genome, read length, biological and technical replications, etc.

Response 3: Thank you for pointing this out. We have supplemented the RNA-seq protocol as required (Pages 7, 128-148).

Comments 4: qPCR protocol (lines 166-178) should be explained indicating statistical analysis, and biological and technical replications.

Response 4: Thank you for pointing this out. We have improved and revised the qPCR protocol as required (Page 8, lines 164-182)

Comments 5: Figures 9a, b, c, d, f, g must be merged in a single Figure.

Response 5: Thank you for pointing this out. We have merged Figures 9A, B, C, D, E, F and G as required (in figure 9 of ALL figure)

Comments 6: Figures 10a, b, c, dg must be merged in a single Figure.

Response 6: Thank you for pointing this out. We have merged Figures 10A, B, C and D as required (in figure 10 of ALL figure)

Comments 7: In Figure 14 authors should be incorporated the correlation coefficient between qPCR and RNA-Seq data in the assayed genes.

Response 7: Thank you for pointing this out. We have conducted Pearson correlation analysis on the transcriptome data and qPCR as required, and marked the correlation coefficient R in the figure (Figure 14 in ALL figure).

Comments 8: Discussion section is limited. Authors must discuss the obtained results with previous data obtained by differ authors.

Response 8: Thank you for pointing this out. We have supplemented and revised the discussion section (page 18, lines 391-443).

Comments 8: Conclusions section (lines 504-514) must be completed with the main implications of the obtained results at bamboo production level in the context of global warming.

Response 9: Thank you for pointing this out. We have supplemented and revised the discussion section (page 20, lines 444-459).

Round 2

Reviewer 2 Report

Comments and Suggestions for Authors

Authorshave revise correctly the manuscript